# A hybrid CNN and ensemble model for COVID-19 lung infection detection on chest CT scans

Ahmed A. Akl[1], Khalid M. Hosny[2]*, Mostafa M. Fouda[3], Ahmad Salah[4,5]

1 Senior Machine Learning Engineer, VA Computing, Cairo, Egypt, 2 Department of Information Technology, Faculty of Computers and Informatics, Zagazig University, Zagazig, Sharkia Egypt, 3 Department of Electrical and Computer Engineering, Idaho State University, Pocatello, ID, United States of America, 4 Department of Computer Science, Faculty of Computers and Informatics, Zagazig University, Zagazig, Sharkia, Egypt, 5 Department of Information Technology, College of Computing and Information Sciences, University of Technology and Applied Sciences, Ibri, Al Dhahirah, Sultanate of Oman

* k_hosny@yahoo.com

**Data Availability Statement:** Data are available at https://www.kaggle.com/plameneduardo/sarscov2-ctscan-dataset.

**Funding:** The author(s) received no specific funding for this work.

## Abstract

COVID-19 is highly infectious and causes acute respiratory disease. Machine learning (ML) and deep learning (DL) models are vital in detecting disease from computerized chest tomography (CT) scans. The DL models outperformed the ML models. For COVID-19 detection from CT scan images, DL models are used as end-to-end models. Thus, the performance of the model is evaluated for the quality of the extracted feature and classification accuracy. There are four contributions included in this work. First, this research is motivated by studying the quality of the extracted feature from the DL by feeding these extracted to an ML model. In other words, we proposed comparing the end-to-end DL model performance against the approach of using DL for feature extraction and ML for the classification of COVID-19 CT scan images. Second, we proposed studying the effect of fusing extracted features from image descriptors, e.g., Scale-Invariant Feature Transform (SIFT), with extracted features from DL models. Third, we proposed a new Convolutional Neural Network (CNN) to be trained from scratch and then compared to the deep transfer learning on the same classification problem. Finally, we studied the performance gap between classic ML models against ensemble learning models. The proposed framework is evaluated using a CT dataset, where the obtained results are evaluated using five different metrics The obtained results revealed that using the proposed CNN model is better than using the well-known DL model for the purpose of feature extraction. Moreover, using a DL model for feature extraction and an ML model for the classification task achieved better results in comparison to using an end-to-end DL model for detecting COVID-19 CT scan images. Of note, the accuracy rate of the former method improved by using ensemble learning models instead of the classic ML models. The proposed method achieved the best accuracy rate of 99.39%.

**Competing interests:** The authors have declared that no competing interests exist.

# 1 Introduction

CT scan images play a crucial role in analyzing and recognizing COVID-19 infections. Radiologists can recognize the COVID-19 patient from a CT scan image with an accuracy rate of 86.27%, as discussed in [1]. The most widely used method for recognizing COVID-19 is the Reverse Transcription Polymerase Chain Reaction (RT-PCR) [2]. This approach searches for the viral RNA from the nasopharyngeal swab. On the other side, the RT-PCR method has two drawbacks. First, its cost is high for residents of developing countries. Second, it is relatively slow. Thus, doctors use chest X-ray and CT scan images as a fast and widely available alternative [3].

It is essential to benefit from the recent advances of DL approaches to identify COVID-19 patients from CT scans. DL-based methods can provide tools with higher accuracy rates than the manual detection approach. In this context, several machine learning methods based on DL CNNs were proposed to diagnose chest X-ray and CT scan images. DL classification models can be classified into two main categories; the first category of methods utilized transfer learning [4], well-known DL approach due to the limited-sized datasets. The second category method builds the DL model by training a CNN from scratch.

The authors in [1] used the transfer learning technique to train ten well-known DL CNN models on a COVID-19 CT scan images dataset. The authors found that the top two CNN models were ResNet-101 and Xception. In [5], the authors proposed using a pre-trained DenseNet-201 to decide if the CT scan image belongs to a COVID-19 patient or not. Similarly, several works depend on the deep transfer learning approach [6–8].

There are two different approaches to building a DL-based image classifier. First, the DL model is built by copying the weights of several layers from an existing trained model and doing additional training to the rest of the model layers to address the problem; this method is called transfer learning. Thus, the DL model has a majority of predetermined weights and a few adjusted/trained weights. Second, The DL model can be built by learning from scratch where all the model weights are completely adjusted/trained. The transfer learning approach is easy to be implemented and fast to train the model, but it should recognize images similar to the images used to train the DL model we transfer its weights. The training from the scratch approach is more suitable when the dataset is new.

In this work, we study the effect of using ready deep learning architectures for predicting the COVID-19 cases from an input CT scan image with the help of the deep transfer learning approach. Thus, most of the weight of the utilized pre-trained deep learning architectures are tuned using images different than CT scan images of COVID-19 patients. Then, we propose using the pre-trained deep learning models to extract the most significant features from CT scan images of COVID-19 patients. Besides, we proposed extracting tract the most significant features using a new proposed DL architecture. These extracted features are then fed to a classic machine learning classifier (e.g., random forest). These two approaches of feature extraction are compared on a widely-used dataset of COVID-19 CT scans. The intuition of the proposed work is that the weights of the pre-trained deep learning models are tuned using images not similar to the CT scan images of COVID-19 patients. Thus, it is better to utilize the deep learning pre-trained models for what is these models good for, which is feature extraction. Then, we train an image classification model from scratch on these significant extracted features.

In this work, we try to answer the question of whether to use the transfer deep learning models for the sake of feature extraction on CT scan images and then use a classic machine learning algorithm for classifying the CT scan images or to use transfer deep learning models for both the feature extraction and classification tasks. We can summarize the motivation of

this work as follows. We propose to investigate the efficiency of the transfer learning approach for COVID-19 CT scan classification in comparison to using the CNN model for feature extraction. Thus, we proposed using a DL CNN model for feature extraction and then using a machine learning classifier (e.g., random forest), which reduces the number of model parameters and the model size. The contributions of this work are listed as follows:

1. We proposed studying the performance of DL as an end-to-end classification model, i.e. deep transfer learning, for CT scans of COVID-19 against using DL models only for feature extraction and then using an ML model for the purpose of image classification.

2. We proposed a new CNN architecture that is trained from scratch. Then, we compared the quality of the extracted features from CT scan images using the well-known DL models (e.g., VGG and MobileNet) against the proposed CNN architecture. The results showed that the proposed CNN architecture extracted features led to higher classification accuracy rates.

3. We compared the results of fusing the extracted features from DL models with the SIFT image descriptor, which resulted in improving the classification accuracy rate. To our knowledge, this is the first study of fusing features extracted by the SIFT method and DL models.

4. We compared the performance of the classic ML methods against the ensemble learning models. The results showed that the ensemble learning models slightly outperformed the classic ML methods. To our our knowledge, this is the first comparison between these two types of methods on this dataset.

5. The proposed methods are evaluated on five different metrics on one of the most used dataset. The obtained results show that the proposed methods outperformed all of the existing state-of-the-art methods. Besides, the ensemble learning-based model slightly outperformed the single classifier.

The rest of the paper is organized as follows. The background is discussed in Section 2. Section 3, we review the performance of the existing methods. Section 4 exposes the proposed method. The empirical results and discussion are exposed in Section 5. Finally, the work is concluded in Section 6.

## 2 Background

Deep learning models that consist of multiple processing layers can learn multiple levels of representations from the data. These models revolutionized several domains such as natural language and text processing, computer vision, computational biology, and more; and achieved state-of-the-art results in these domains surpassing the human perception [9]. The most common form of deep learning is supervised learning, where you need to have a large dataset of images [10]; each image is labeled with its category, e.g., image classification. One of the most prominent models is Convolutional Neural Networks (CNNs). CNNs are developed to process data in multiple dimensions, e.g., images and videos [11]. The main layers that compose the CNN model are Convolutional layers, Pooling layers (e.g., Max Pooling, and Fully Connected layer). The Convolutional layer is used for feature extraction, the pooling layer for dimensionality reduction, and the fully connected layer for the classification task. Deep convolutional neural networks perform well in feature extraction. The deep the model is the more abstract features extracted as early convolutional layers capture low-level features from the images. Unfortunately, increasing the CNN model depth requires more labeled data which is

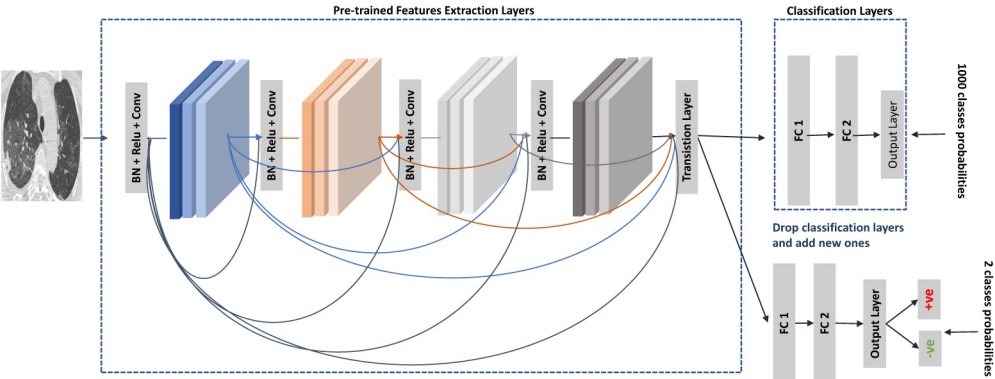

**Fig 1. Transfer learning example: DenseNet201 pre-trained feature extraction layers.**

the main drawback of training CNN models. Collecting enough training data is time-consuming, unrealistic in some cases, and expensive.

Transfer deep learning was introduced to overcome the limitations in real-world problems and train the CNN model on small labeled data. The main idea of Transfer Learning is transferring the knowledge across different but related domains [12]. The CNN model is pre-trained on a large dataset, e.g., ImageNet [13], containing images for 1,000 categories and fine-tune it into a new task with two or three categories. Thus we will freeze all the CNN layers that extracted pre-trained features from the original dataset, dropping the classification layer and adding the new classification layer. Transfer Learning is applied in many domains and achieves superior results. In [5], the authors applied deep learning transfer learning to classify CT images for the COVID-19 classification task in the medical domain. As shown in Fig 1, we can use DenseNet201 pre-trained convolutional layers as feature extraction layers to extract the essential features from the CT images, then feed them into a classification layer to decide if the image is positive or negative.

## 3 Related work

There is massive number of research works conducted to address the task of COVID-19 detection from the CT scan images and chest X-ray [14, 15]. These methods can be classified by the utilized approach (e.g., machine learning or deep learning). Recently, deep neural networks have been applied successfully in image classification tasks and achieved outstanding results. Several efforts utilized the existing DL model (e.g., ResNet-101 and DenseNet-201). For instance, in [1], the authors proposed using ten widely used CNN architectures to predict infection of COVID-19 patients from CT scan images. The reported results of the ten models outlined the performance gaps on different evaluation metrics such as sensitivity and accuracy. In the following, we organize the existing efforts to classify COVID-19 patients from CT scan images using the CNN model (e.g., MobileNetV2 and Xception). The complexity of the CNN model varied from one architecture to another, as shown in Table 1. The more the number of parameters, the more the model storage size and running time. We will review and discuss recent methods applied to detect COVID-19 from the CT scan images based on the architecture proposed and expose the advantages and limitations of these methods.

DenseNet-201 is a CNN consisting of 201 layers. DenseNet-201 uses the preceding layers as each layer obtains input from the entire previous layers and sends forward its own extracted features to the subsequent layers, which improves the performance [16]. In [5], the authors proposed using DenseNet-201 with an extra three fully connected layers to classify the public

**Table 1. Comparison of the well-known CNN models used in deep transfer learning on the number of parameters.**

| CNN Model | No. of parameters |
| --- | --- |
| Inception V3 | 23.9M |
| VGG-19 | 138M |
| AlexNet | 61M |
| ResNet-50 | 23M |
| DenseNet-201 | 20M |
| MobileNetV2 | 3.4M |

COVID-19 dataset in [17]. Their proposed model was able to classify the chest CT-scans images with a training accuracy of 99.82%, testing accuracy of 96.25%, and validation accuracy of 97.4%, respectively. The main drawback of this model is that it has a an overfitting issue, as the difference between the training and test accuracy scores is 3.75%. In addition, the authors in [18] proposed a framework by utilizing the pre-trained DenseNet-201 DL model in two ways one for features extraction and the other was retrained using transfer deep learning. Then, the extracted features were screened by using the most significant features with the help of a Firefly optimization algorithm. This proposed method has two limitations the first is optimal features selection using the Firefly algorithm that might not select the most relevant feature and sometimes reject the most important features. The second is computational cost due to feature fusing.

ResNet [19] is proposed to address the vanishing gradient problem in the neural networks. ResNet's main idea is to skip updated layers and then connect these layers directly to the output. A variant of ResNet is ResNet-50; ResNet-50 consists of 50 layers. The authors in [20] utilized the ResNet-50 architecture to classify chest CT scans of COVID-19 and normal cases. In [21], the authors proposed FCONet; it is a hybrid model of a straightforward 2D DL framework that used only a single chest CT image. This model provides superb diagnostic results in detecting COVID-19 pneumonia. Based on the evaluation experiments, the FCONet model based on ResNet-50 architecture outperformed other FCONet models utilizing both Inception-v3 and Xception architectures. The limitation of their proposed model is the overfitting issue, as all the dataset was obtained from the same source the model can not generalize well.

VGG [22] is built over an improvement on AlexNet. This improvement includes reducing the kernel-sized filters with multiple 3 × 3 kernel-sized filters instead of 11 and 5 in the first and second convolutional layers. In [23], the authors proposed an attention-based architecture; the author's utilized VGG-16 architecture and the attention module. The role of the attention module is to recognize the spatial relationship between the regions of interest in x-ray images. The authors did not utilize offline data augmentation like GANs or Convolutional Autoencoder before training may decrease the overfitting problem. Also, They did not use other pre-trained deep learning models with smaller filter size which could increase the performance. Using X-ray and CT scan chest images, the authors in [24] introduced a CNN model to recognize COVID-19 cases. The evaluation of the introduced methods show that the VGG-16 model has the best accuracy rate followed by Inception-V2. Finally, the authors compared the results with a simple decision tree model. The utilized dataset is small-size; it includes only 394 images. Thus, their proposed models suffer from overfitting issues, especially the Inception-V2 model. Similarly, the authors in [25] proposed tuning the VGG19 model on a small dataset. The obtained accuracy of classifying the CT scan image of the proposed method is only 84% while other deep learning models perform poorly.

Sandler et al. [26] proposed the MobileNetV2 model as a CNN architecture for machines with limited computing power, like smartphones. The MobileNet architecture compromises

accuracy and memory requirements. The main idea of MobileNet architecture is to decrease the number of learning parameters. In [27], the authors addressed the problem of detecting COVID-19 cases from X-ray images using several well-known CNN models. The Mobile-NetV2 has a modest accuracy performance, as the MobileNetV2 model sacrifices accuracy in cost of reducing the model memory space (i.e., model size). The main demerit of this work is using a dataset consists of only 50 X-ray images divided into two classes, positive and negative, which led to overfitting problem.

Inception V3 is a CNN architecture that provides several enhancements that include utilizing label smoothing, factorized $7 \times 7$ convolutions, and an additional classifier to pass label information lowers down the network [28]. In [29], the authors proposed an automated classification method of COVID-19 from X-ray images. The authors utilized a transfer learning technique by utilizing CNN-based models such as Inception-v3 and ResNet-50. The reported results outlined that Inception V3 slightly outperformed ResNet-50 is in the process of COVID-19 recognition from X-ray images. The reported accuracy rates are 99.01% and 98.03% for the pre-trained Inception V3 and the ResNet-50 models, respectively.

Several methods utilize feature selection or fusing the extracted features from several well-known CNN models. In this context, the authors in [30] proposed a two-stage end-to-end deep learning framework for COVID-19 detection. The first stage is feature extraction by utilizing the following state-of-the-art ResNet152, DenseNet201, and Xception models. The second stage is feature selection they used Harmony Search (HS) and Adaptive B-Hill Climbing (ABHC) algorithms to filter the insignificant features from the feature vectors. The proposed framework achieved best scores with HS and local search via ABHC of 97.30% and 98.87% respectively. The main limitation is that the model can not be able to classify the case as positive in the early stages of the infection. In addition, in [31], the authors proposed a new framework to classify COVID-19 lung patients' images. The system prepossess the images then use two types of feature extraction; deep learning and handcrafted methods. The features extracted from both models are fused for the feature selection process. The authors in [32] proposed a framework that utilized one of the pre-trained models, DenseNet, ResNet, VGGNet, or Mobi-leNet, and capsule network (CapsNet) to classify the CT-scan images. The experiments found that THE CapsNet achieves the best results with lighter deep neural architecture. Finally, the selected features forwarded to the classification phase. In [33], the authors proposed merging the features of the following pre-trained network architectures AlexNet [11], SonoNet64 [34], XNet (Xception) [35], and InceptionV4 [36].

In [37], the authors proposed utilizing different DL models to extract features from the input CT scan image and then several ML model were utilized to classify the input CT scan image based on the extracted feature with the DL model. The limitations of this work are three-fold. First, the authors utilized a small dataset, 349 CT scan images for the COVID-19 class and 397 CT scan images for the non-COVID19 class. Second, the authors did not explore the training from the scratch approach for feature extraction of COVID-19 classification datasets. Third, the authors did not compare their proposed methods with the state-of-the-art methods. Moreover, the authors did not fuse the features extracted by the DL model with classic feature extraction such as SIFT.

It can be concluded that there are several efforts achieved to detect COVID-19 patients from X-ray or CT scan images. These efforts use different datasets; thus, it is difficult to assess the best-performing methods. On the other side, all of these efforts followed the deep transfer learning approach, which motivates this work to investigate the other approach of training a CNN model from scratch and compare these two approaches. Besides, the second motivation was to investigate the performance gap between using the deep transfer learning as a classifier

against using it as a feature extracting method, as most of the literature utilized the deep transfer learning approach as end-to-end models.

# 4 Materials and methodology

## 4.1 Dataset

The SARS-CoV-2 (https://www.kaggle.com/plameneduardo/sarscov2-ctscan-dataset, last accessed: 24 Feburary, 2023) CT scan dataset [17] is publicly available on Kaggle for research and experiments. According to the authors of the dataset, they obtained all the required patient/participant consent. In addition, they have been obtained and archived the institutional forms to approve this dataset. The utilized dataset consists of two classes for positive and negative SARS-CoV-2 infections, as shown in Fig 2, where the positive class has 1,252 samples and the negative class has 1,230 samples. The dataset consists of 2,482 CT scan images. we divided the entire dataset into three groups; train, validation, and test with percentages 70%, 10%, and 20%, respectively. The training set is used to learn and tune the model weights. The validation set is used to validate the model performance during the training process. The test set (i.e., unseen samples) is utilized to measure the model performance. We did not use any augmentations methods to make the task more challenging by using this limited data, we just normalized the images.

## 4.2 Implementation details

All of the performed experiments are written in the Python programming language. The experiments are preformed on a computer with Linux OS with a dual core Processor Intel(R) Core(TM) of 2.50GHz, 24GB RAM. We divided the entire dataset into three parts: training, validation, and testing to overcome the overfitting problem and make the model able to generalize well. In addition, we utilized Reduce learning rate On plateau and Early stopping methods that help in the generalization task. The input images are pre-processed by resizing and normalize them into 224 × 224 × 3 to fit the model's input shape. All of the proposed deep learning models are built using Tensorflow 2.5, the well-known DL framework.

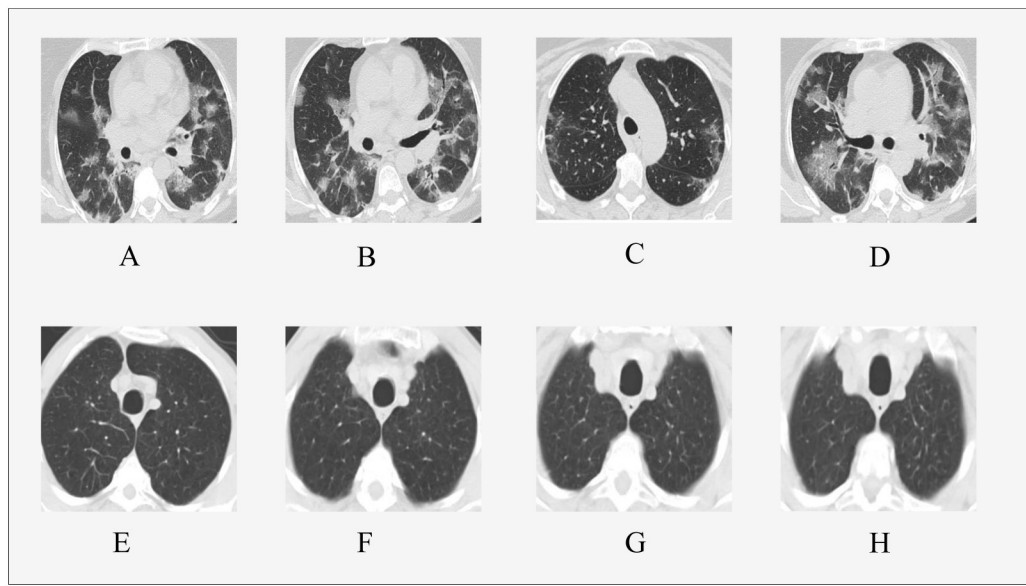

**Fig 2. A sample of CT-scan images.** (A-D) COVID-19 positive cases and (E-H) COVID-19 negative cases.

We validate the proposed method performance by comparing it with the state-of-the-art DL models for COVID-19 CT scan image classification. All other methods used deep transfer learning (e.g., VGG16, DenseNet) for this classification task. The evaluation of the proposed model consists of four points. First, we compared the proposed method against the state-of-the-art methods on several accuracy metrics (e.g., precision, recall, and F1-score). The second point of comparison is the ROC AUC curve to understand the performance of the image classification model against the no-skill model. The third point of comparison is the confusion matrix for the proposed method to outline the performance of the proposed models for the two classes. Finally, the fourth last point of comparison is the overfitting analysis of the proposed model.

### 4.3. The proposed method

The proposed framework includes two main phases for COVID-19 detection. In the first phase, we extract the most significant features from the CT scan images. We tested individually three major methods for the feature extraction: 1) the pre-trained deep learning models, 2) a newly proposed CNN architecture, 3) SIFT algorithm. Then, the best two feature extraction methods are combined together to generate the features vector for the classification phase. As shown in Fig 3, we have selected both the proposed CNN model and SIFT as feature extraction. The second phase is the classification process; we utilized these extracted features as input to classic machine learning classifiers such as Random Forest and Support Vector Machines.

**4.3.1 Feature extraction phase.** We utilized three approaches for the feature extraction phase, namely, transfer learning, training from scratch, and image descriptor methods, e.g., SIFT. In the transfer learning approach, we proposed training the MobileNetV2 architecture on the COVID-19 dataset to learn the discriminative features of positive and negative classes. Then, we utilized this fine-tuned network as a feature extractor. The extracted features of COVID-19 images are passed into a machine learning classifier model, which decides whether the input image is a COVID-19 positive or negative.

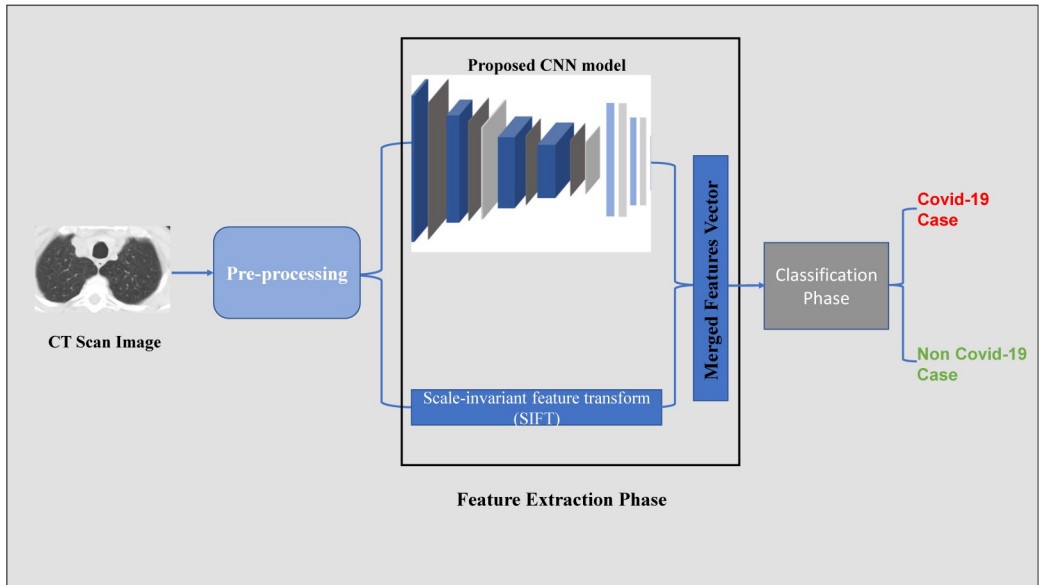

**Fig 3. The proposed framework for COVID-19 detection.**

MobileNetV2 model utilizes depth-separable convolution with residuals as the basic building block. The architecture of the MobileNetV2 model achieves state-of-the-art results in the classification task on the ImageNet dataset [11] compared to ShuffleNet [38], and Mobile-NetV1 [39]. In MobileNetV2 architecture, the last convolution layer output feature map with the shape $7 \times 7 \times 1,280$ which are 62,720 features. We handled these features in two different scenarios. First, these features will be used as input to the classifier directly, in the classification phase. Second, we proposed fine-tuning the model weights on top of these features consist of a Convolutional-2D layer with 64 filters, where the kernel size is $3 \times 3$. The output of the mentioned Convolutional-2D layer is an activation map with $5 \times 5 \times 64$ followed by a MaxPool-2D layer with kernel size $2 \times 2$ that produces an output of shape $2 \times 2 \times 64$. Then, we flatten this feature map and add a dense layer with 128 neurons. The final output of the model is 128 numerical values, and these values are considered the extracted features. Thus, these 128 features will be consumed as the input to the classifier in the classification phase.

The second approach is to train a CNN model from scratch; we proposed designing a new CNN architecture and training it from scratch to adjust the weights of the entire model. Training a model from scratch will capture all the discriminative features of the dataset. Fig 4 depicts the architecture of the newly introduced model. The proposed CNN model consists of 17 layers. The activation function of all of the layers is the Swish function, and it is a smooth, non-monotonic function. Its equation is illustrated in Eq 1.

Several useful reports show that the Swish activation function works better than ReLU across several challenging datasets, encouraging our proposed model to utilize the Swish activation function. The first layer is an input convolutional-2D layer. The input shape of this layer is $224 \times 224$ and the depth of this layer is three, as the colored image has three channels. We used 128 filters with a filter size $3 \times 3$ which produces a feature map with shape $222 \times 222 \times 128$. Every Convolution layer is followed by a pooling layer with kernel size $2 \times 2$; to reduce the feature map dimension to nearly half, we utilized Max pooling as its practical achievements.

Second, the input layer is followed by three convolutional- 2D layers, where every layer uses 256 filters with a kernel size of $3 \times 3$. These convolutional-2D layers are separated by Max-pool2D layers filter size $2 \times 2$. We dropped out after the second and the fourth Maxpool2D layers with rates of 0.25 and 0.20, respectively; that helps the model to fight the overfitting problem. These blocks of Convolutional, Maxpooling, and Dropout layers are used to extract abstract features from the input image. Finally, the model contains a set of three dense layers; a

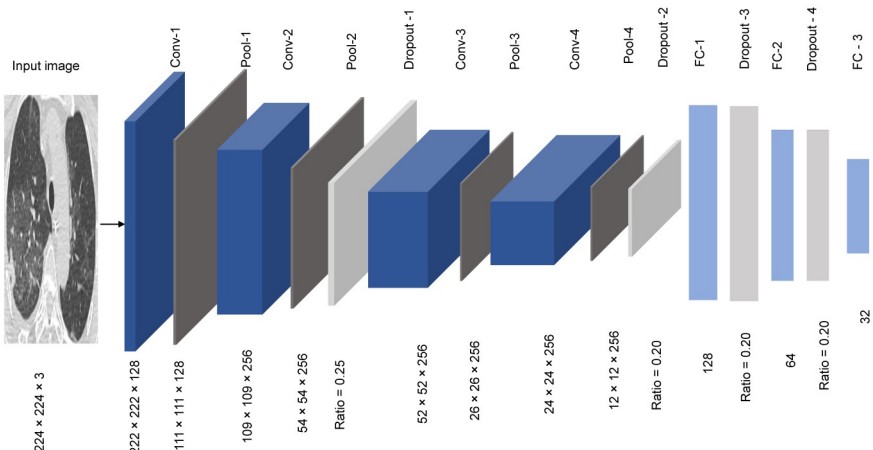

**Fig 4. The proposed CNN model for feature extraction.**

dropout separates these layers with a ratio equal to 0.2. These three dense layers sizes are 128, 64, and 32, respectively. The last dense layer size represents the size of extracted features. Thus, each CT scan image can be represented by 32 features. These features will be fed to the classifier to decide whether these features represent a COVID-19 case or not.

Because we are dealing with a small dataset, we encouraged to use Scale-Invariant Feature Transform (SIFT) [40] algorithm to extract the key features from the images. The extracted features with SIFT will be combined with the extracted features from proposed CNN model. The final combined features will be passed to the classification phase. We noticed that utilizing SIFT algorithm slightly increased the classification model's performance.

$$f(x) = x \times sigmoid(x) \tag{1}$$

**4.3.2 Classification phase.** For feature extraction task, we utilized the proposed CNN architecture, SIFT and MobileNetV2 model. We utilized machine learning classifiers to classify the extracted features as COVID-19 case or not. The machine learning classifier consumes numerical values, as extracted from the CNN model, rather than a CT scan image. We proposed comparing the performance of a single classifier and an ensemble learning classifier by classifying the extracted features from the feature extraction phase. We utilized random forest, an example for Bagging ensemble methods, which depends on a decision trees model as a base classifier. Boosting methods are applied also by utilizing a support vector machine (SVM) classifier as a weak learner. On the other hand, a single classifier, logistic regression, is used to classify the extracted features in the previous stage.

The standard MobileNetV2 model extracts 62,720 numerical values with shape $7 \times 7 \times 1,280$ of the CT scan image; these features are used as an input to the classification models. MobileNetV2 model with the fine-tuned model on top of it extracts 128 numerical values of the CT scan image for the classification process, as listed in Table 2. The proposed CNN model extracts 32 features from the CT scan image that are used as input for the classification task. As the proposed CNN model achieves better results than the pre-trained MobileNetV2 model, we were encouraged to merge the features extracted from proposed CNN and SIFT to together. As listed in Table 2, utilizing SIFT features with our proposed CNN features achieves the superior results.

Several metrics can evaluate the performance of a classifier. In the following, we explain the utilized metrics for evaluating the proposed classifier. Besides, we mean by True Positive (TP) the outcome where the model correctly predicts the positive class, e.g., positive COVID-19 patients diagnosed as COVID-19 (+). A True Negative (TN) is a results where the model predicts the negative class flawlessly, e.g., a CT scan image of COVID-19 patient is diagnosed as a negative COVID-19 (-) case. A False Positive (FP) case where the model incorrectly forecasts

**Table 2. Testing scores of the proposed methods for COVID-19 classification.**

| Model | Precision | Recall | F1-score | Specificity | Accuracy MobileNetV2 |
|---|---|---|---|---|---|
| MobileNetV2 (Transfer Learning) | 96.70% | 97.00% | 96.70% | 96.80% | 96.80% |
| MobileNetV2 (Feature Extraction) + RF | 98.40% | 99.20% | 98.80% | 98.40% | 98.79% |
| Proposed model+LR | 98.80% | 99.20% | 99.00% | 98.8% | 98.99% |
| Proposed model + RF | 98.80% | 99.20% | 99.00% | 98.80% | 98.99% |
| SIFT + proposed model + LR | **99.20%** | 99.20% | 99.20% | 99.20% | 99.19% |
| SIFT+proposed model+Bagging Method (SVM) | **99.20%** | **99.60%** | **99.40%** | **99.20%** | **99.39%** |
| SIFT+proposed model+Boosting Method (SVM) | **99.20%** | 99.20% | 99.20% | 99.20% | 99.19% |

the positive class, e.g., patients suffering from other lung diseases and incorrectly classified as COVID-19 (+). False Negative is a case where the model incorrectly forecasts the negative class, e.g., patients infected by COVID-19 (+) and incorrectly classified as COVID-19 (-). Accuracy is defined as the fraction of correct predictions (both True Positive and True Negative).

The first evaluation metric is precision; where can it can be defined as the rate of TP outcomes to the total number of positive outcomes (TP + FP), as shown in Eq 2.

$$Precision = \frac{TP}{TP + FP} \tag{2}$$

The second evaluation metric is the recall; it is the ratio of TP outcomes to the actual number of positive samples (TP + FN), as shown in Eq 3.

$$Recall = \frac{TP}{TP + FN} \tag{3}$$

The third evaluation metric is the F1-score; it is the harmonic mean of precision and recall, as shown in Eq 4.

$$F1 - score = \frac{TP}{TP + 0.5(FP + FN)} \tag{4}$$

The fourth evaluation metric is the specificity; It is the ratio of TN outcomes to the total negative outcomes (TP + FN) of the model, as shown in Eq 5.

$$Specificity = \frac{TN}{TN + FP} \tag{5}$$

The fifth evaluation metric is the accuracy; It is the ratio of correct outcomes (TP + TN) to the total number of outcomes, as shown in Eq 6.

$$Accuracy = \frac{TP + TN}{TP + TN + FP + FN} \tag{6}$$

## 5 Results and discussion

### 5.1 Results

We conducted a set of experiments to select the best architecture for the proposed framework. Five performance metrics are used for the selection process. Table 2 shows the results of the proposed methods for both the features extraction and classification phases. We can see that features extraction with both SIFT and the proposed CNN model then classify these features with bagging ensemble method achieves the best results.

The optimal architecture is compared with the state-of-the-art DL models using various performance metrics. The most common metrics used for this case are precision, recall, F1-score, specificity, and accuracy. Table 3 lists the scores of the five evaluation metrics for the proposed framework and the other state-of-the-art methods. The proposed method achieved the highest metrics scores.

The second point of comparison is the ROC AUC curve for evaluating the performance of utilizing the MobilenetV2 model as a transfer learning approach for feature extraction against using the proposed CNN-based model for the same purpose. Fig 5 depicts the ROC AUC curve of the three proposed methods. Fig 5(a) and 5(b) show close to perfect results in

**Table 3. Comparison of the proposed model and the state-of-the-art DL-based models for COVID-19 classification.**

| Model | Precision | Recall | F1-score | Specificity | Accuracy |
|---|---|---|---|---|---|
| VGG16 [5] | 95.74% | 95.23% | 95.49% | 95.67% | 95.45% |
| Inception ResNet [5] | 90.15% | 92.06% | 91.09% | 89.72% | 90.90% |
| Resnet-152 V2 [5] | 92.92% | 97.35% | 95.09% | 92.43% | 94.91% |
| DenseNet [5] | 96.29% | 96.29% | 96.29% | 96.21% | 96.25% |
| GLCM+encoder [41] | 97.77% | 98.77% | 97.78% | 96.78% | 97.78% |
| xDNN [17] | 99.16% | 95.53% | 97.31% | – | 97.38% |
| **SIFT+proposed model+Bagging Method (SVM)** | **99.20%** | **99.60%** | **99.40%** | **99.20%** | **99.39%** |
| [42] | - | - | 85.00% | - | 86.00% |
| CNNs+SVM [43] | 93.0% | – | 94.0% | 93.7% | 94.70% |
| [44] | 80.52% | – | 83.99% | 88.24% | 84.57% |
| [45] | – | – | 85.00% | – | 83.00% |
| ResNet50 [46] | – | – | – | 91.43% | 82.64% |
| DenseNet201 + HS + ABHC [30] | – | – | – | – | 97.30% |
| MobileCapsNet [32] | 97 | – | 99 | – | 99% |

comparison to the no-skill model. Fig 5c depicts the ROC AUC curve of the proposed model, which uses the MobileNetV2 as feature extraction and the RF as a classifier; this model slightly performed less accurately than the two models above.

Fig 6 depicts the precision-recall curves of the three proposed methods. Fig 6(a) and 6(b) show high precision and recall rates result in comparison to the no-skill model. Fig 6(c) depicts the precision and recall curve of the introduced model which uses the MobileNetV2 as feature extraction and the RF as a classifier; this model slightly performed less accurate than the afore-mentioned two models.

Fig 7 depicts the confusion matrices of the three proposed methods. Fig 7(a) and (7b) show high TP and TN rates. Fig 7(c) depicts the confusion matrix of the introduced model which uses the MobileNetV2 as feature extractor and the RF as a classifier; this model slightly has lower TP and TN rates than the two models.

Fig 8 shows the overfitting analysis for the MobileNetV2 model. There is a clear overfitting as shown in Fig 8(a) and 8(b), because the difference between the training and test scores is larger than the accepted threshold. On the other hand, the overfitting analysis of the proposed methods shows no significant difference, less than 0.001, between the training and test accuracy and loss scores.

## 5.2 Discussion

The exposed results revealed several points of interest and answer the research question of this work. The first point of interest compare the performance of the pre-trained DL models (e.g., MobileNetV2) as feature extractor against building a customized DL model which trained from scratch on COVID-19 lung CT scan images. The experimental results show that the proposed model which utilized the swish activation function achieved slightly better performance, as shown in Table 2. The proposed method results, which is training a DL model from scratch for feature extraction and the results of using the pre-trained DL models for feature extraction are listed in Table 2. Thus, training a DL model from scratch should yield a better quality of the extracted features. This can be justified by the fact that the pre-trained models are trained on several types of images (e.g., car, building, and lung scans images.), while the approach of training a DL classifier model from scratch focuses on training the DL model on only lung CT

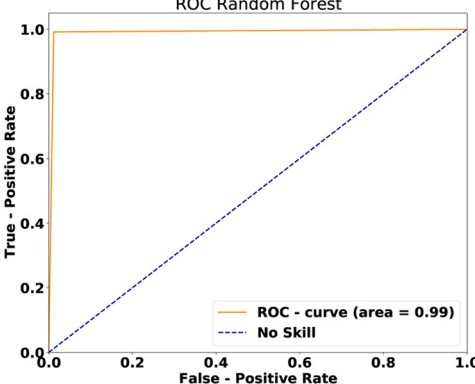

(a) The proposed CNN-based model for feature extraction + random forest classifier.

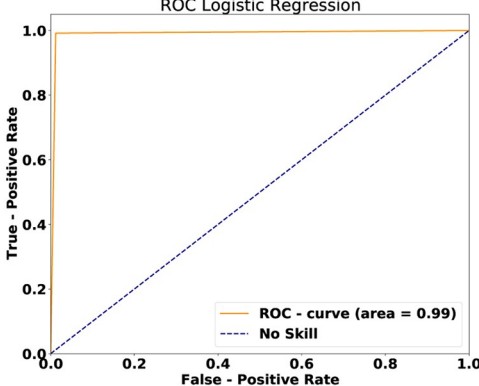

(b) The proposed CNN-based model for feature extraction + logistic regression classifier.

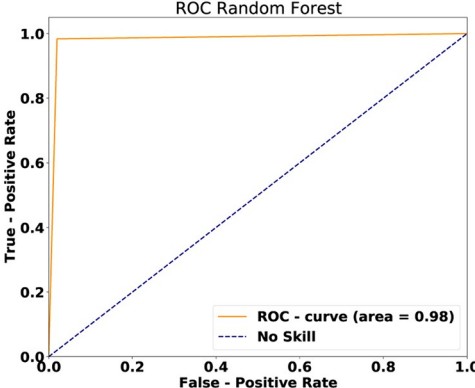

(c) Tuned MobileNetV2 architecture + random forest classifier.

**Fig 5. ROC AUC curve (a) the proposed CNN model+random forest classifier (b) the proposed CNN model +logistic regression classifier (c) tuned MobileNetV2 + random forest classifier.**

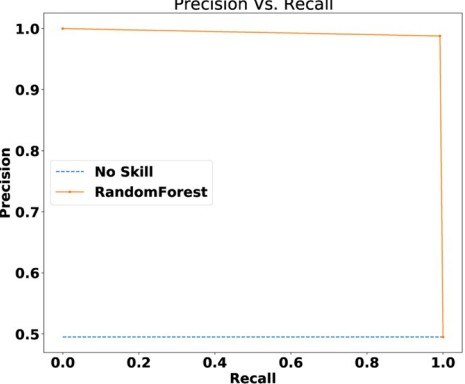

(a) The proposed CNN-based model for feature extraction + random forest classifier.

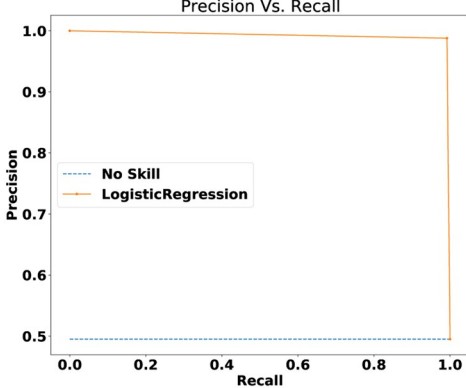

(b) The proposed CNN-based model for feature extraction + logistic regression classifier.

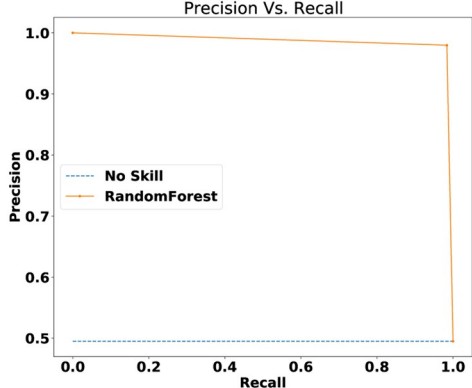

(c) Tuned MobileNetV2 architecture + random forest classifier.

**Fig 6. Precision-Recall curves (a) the proposed CNN model+random forest classifier (b) the proposed CNN model +logistic regression classifier (c) tuned MobileNetV2 + random forest classifier.**

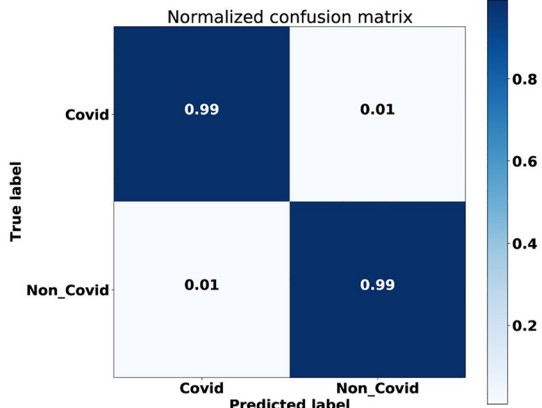

(a) The proposed CNN CNN-based model for feature extraction + random forest classifier.

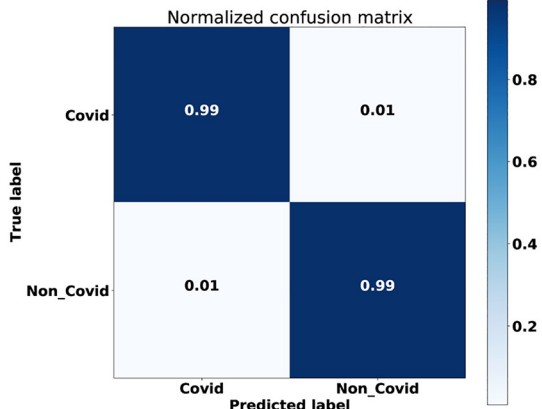

(b) The proposed CNN-based model for feature extraction + logistic regression classifier.

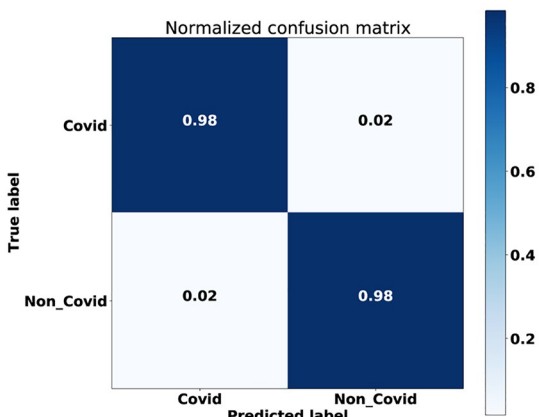

(c) Tuned MobileNetV2 architecture + random forest classifier.

**Fig 7. Confusion matrices (a) the proposed CNN model+random forest classifier (b) the proposed CNN model +logistic regression classifier (c) tuned MobileNetV2 + random forest classifier.**

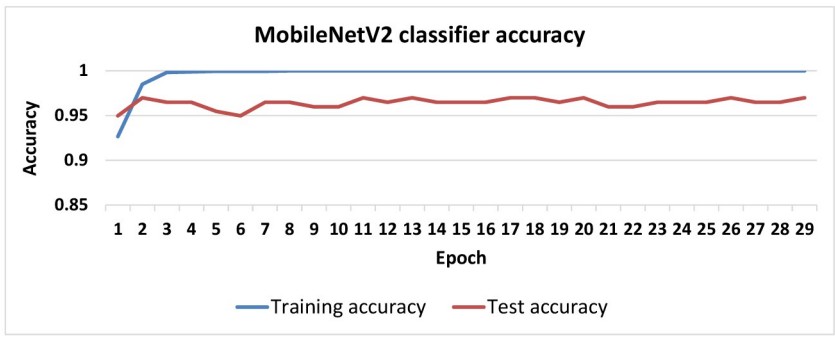

(a) Accuracy.

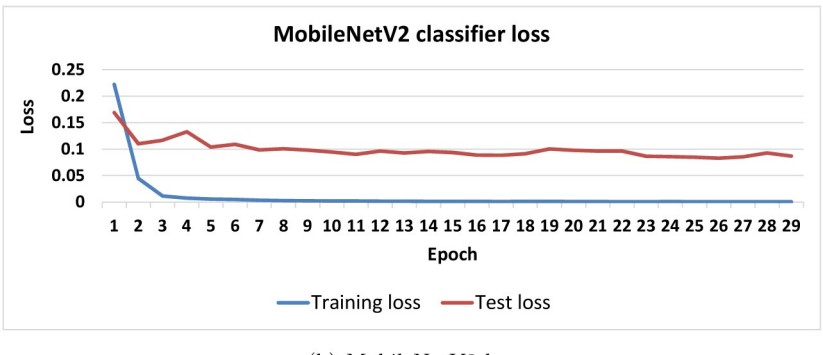

(b) MobileNetV2 loss.

**Fig 8. Evaluation metrics of tuned MobileNetV2 architecture (a) Training and test accuracy (b) Training and test loss.**

scan images with normal and COVID-19 labels. Thus, training the DL model with specialized images might give better results.

The second point of interest addressed the first research question. At this point of interest, we compared two different approaches, namely, using a DL model only and using DL and classic machine learning for classifying images of infect lung with COVID-19. In other words, the first approach uses a end-to-end DL model for the feature extraction and classification tasks, while the second approach uses the a DL model for feature extraction and then uses a classic machine learning model for the classification task. As shown in Table 2, the latter outperformed all of the other methods of comparison for the overall performance on the utilized five metrics. The results of the methods used DL models for features extraction and classification are listed in the 1st to the 6th rows and the 10th to 12th rows in Table 3. On the other hand, the results of the proposed methods, which combined a DL model with a classic machine learning model, are listed in Table 2.

In Table 3, some methods of comparison have no scores for certain metrics; this is linked to the fact that the authors did not provide their source code to reproduce their results nor did they mention the value of the missing metric(s) in their paper. Thus, we can summarize the first point of comparison as follows. Using the transfer deep learning models for the sake of feature extraction on CT scan images and then using classic machine learning algorithm for the sake of the classification tasks achieved better results rather than using transfer deep learning models for both the feature extraction and classification tasks. This can be linked to the fact that the well-known deep learning methods are trained on images different than the CT scan images. Of note, the results of Fig 8 shows that the deep transfer learning model (i.e.,

MobileNetV2) suffers from the overfitting, while the proposed method has no overfitting problem.

For the visual results, Fig 5 shows that there is a slight performance gap between using a pre-trained DL model and to re-train a DL model, where the latter is better. The re-train a DL model (i.e., Fig 5(a) and 5(b)) from scratch was better than the pre-trained models (i.e., Fig 5 (c)) by 1%. The ROC AUC results were 98% and 99% for the former and latter methods, respectively. These results are emphasized in Figs 6 and 7 as well.

The final point to be discussed is the performance of using a classic machine learning classifier (e.g., Logistic Regression) and using the ensemble learning (e.g., a set of weak classifiers such as bagging method). The listed results in Table 2 show that the ensemble learning results (i.e., rows 5 to 7) slightly better than the strong classifier's results (i.e., rows 2 to 4). In addition, combining the extracted features of the proposed DL model with the SIFT extracted features has no effect on the precision, recall, and specificity metrics, while slightly improving the scores of F1-score and accuracy metrics. The reported results based on using a re-trained model only for feature extraction are reported in the rows 3 and 4, and the results of the combined features of SIFT and DL model are reported in rows from 5 to 7 of Table 2.

Model robustness is used to evaluate the model's performance if the input data changes. In our case, the data automatically produced using computed tomography scanners. Thus, we applied the cross-validation technique to test the model's robustness regarding data change.

The cross-validation test was performed for 10 folds for the model with the best performance, the 6th row of Table 2. The mean and accuracy rate of the 10 folds is 98.60% and ±1.96%, respectively. The trust level of the proposed method is computed with the 95% confidence interval. For the same 10 folds, the best performing model has 98.60% with ±1.04% error margin with 95% confidence interval, as depicted in Fig 9.

The limitations of this study are two-fold. First, the performance of the ensemble DL models is not explored. As a single DL model outperformed a single ML model in a general image

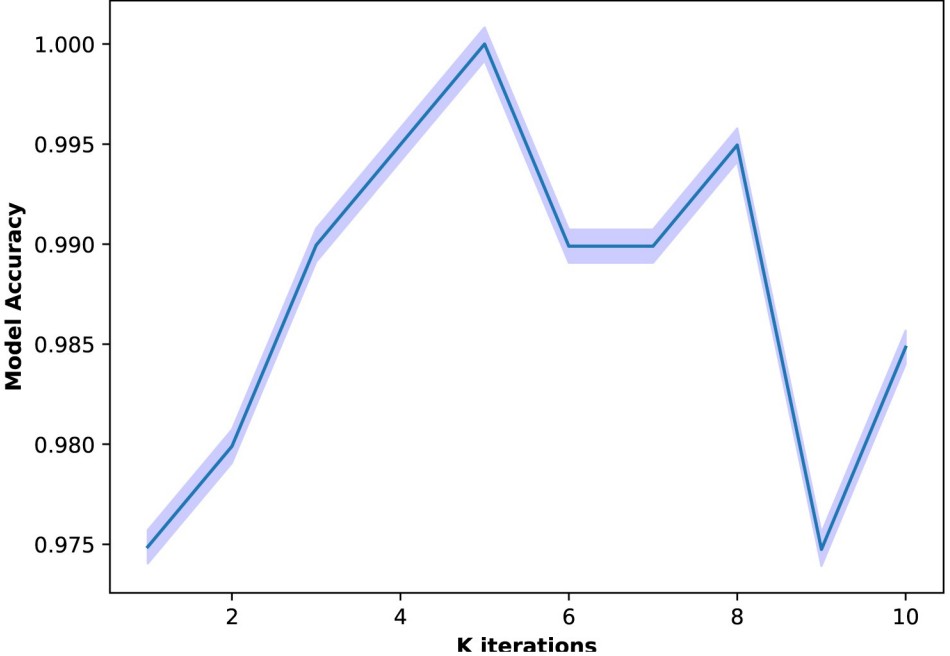

**Fig 9. The proposed model's accuracy with 95% confidence interval for 10 cross-validation folds.**

classification problem, it is expected that the ensemble DL model approach achieve better results than the proposed method. Second, the dataset size is 2,482 images. A dataset with larger number of images can better prove the proposed methods.

## 6 Conclusion

The detection of COVID-19 cases from CT chest scan images is a vital task in the current pandemic. The mainstream for this task is to utilize end-to-end deep learning models, while the quality of the extracted features using DL models is not well studied. Thus, we proposed evaluating the DL models as feature extraction methods for COVID-19 detection based on CT scan images. Thus, several well-known DL models such as VGG and ResNet are utilized to extract features from a COVID-19 dataset. Then, the extracted features are used to classify an input image as a normal case or COVID-19 case using an ML method. In addition, we proposed a new CNN model to compare the transfer DL approach against the training from scratch on the feature extraction task. Moreover, the classification machine learning models are compared against the ensemble learning models for the classification task. The experimental results of the utilized dataset revealed that it is better to use DL models for feature extraction rather than as end-to-end models. The ensemble learning model slightly outperformed the classic ML models. Finally, the proposed CNN model produced better-extracted features in comparison to the well-known DL models. The future direction includes comparing the performance of the ensemble DL models against the ensemble ML models.

## Author Contributions

**Conceptualization:** Khalid M. Hosny, Ahmad Salah.

**Data curation:** Ahmed A. Akl, Khalid M. Hosny.

**Formal analysis:** Ahmad Salah.

**Investigation:** Mostafa M. Fouda, Ahmad Salah.

**Methodology:** Khalid M. Hosny, Ahmad Salah.

**Resources:** Khalid M. Hosny.

**Software:** Ahmed A. Akl.

**Supervision:** Khalid M. Hosny, Mostafa M. Fouda.

**Visualization:** Ahmed A. Akl.

**Writing – original draft:** Khalid M. Hosny, Ahmad Salah.

**Writing – review & editing:** Khalid M. Hosny, Mostafa M. Fouda, Ahmad Salah.

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
