## [Decision Letter · Decision Letter 0]

11 Apr 2022

PONE-D-21-38730A Hybrid CNN and Random Forest Model for COVID-19 Lung Infection Detection on Chest CT ScansPLOS ONE

Dear Dr. Hosny,

Thank you for submitting your manuscript to PLOS ONE. After careful consideration, we feel that it has merit but does not fully meet PLOS ONE’s publication criteria as it currently stands. Therefore, we invite you to submit a revised version of the manuscript that addresses the points raised during the review process.

We look forward to receiving your revised manuscript.

Kind regards,

Le Hoang Son, Ph.D

Academic Editor

PLOS ONE

2. Please ensure that you refer to Figure 1 in your text as, if accepted, production will need this reference to link the reader to the figure.

**Comments to the Author**

1. Is the manuscript technically sound, and do the data support the conclusions?

Reviewer #1: Partly

2. Has the statistical analysis been performed appropriately and rigorously? 

Reviewer #1: N/A

3. Have the authors made all data underlying the findings in their manuscript fully available?

Reviewer #1: Yes

4. Is the manuscript presented in an intelligible fashion and written in standard English?

Reviewer #1: Yes

5. Review Comments to the Author

**Editor:**

- Authors are requested to further enhance the proposed method with more techniques.

- Comparison against the state-of-the-arts should be done.

**Reviewer #1**: 

The paper proposes a comparison between two approaches for predicting COVID-19 patients based on the CT scan images. The first approach uses deep transfer learning to classify the images, the second one is based on CNN to extract features from the images and classify them using ramdom forest algorithm.

The paper is well written and easy to understand. The problems being addressed are interesting. However, the paper lacked novelty. The authors used deep learning techniques that are well known and widely used in medical image classification.

I invite the authors to use more innovative methods and present in-depth the technical details.

---

## [Author Response · Author response to Decision Letter 0]

6 Jun 2022

PLOS ONE

REPLY TO COMMENTS

Ref. No.: PONE-D-21-38730

Paper Title: A Hybrid CNN and Random Forest Model for COVID-19 Lung Infection Detection on Chest CT Scans.

Dear Editors and Reviewers,

Thank you for your useful comments and feedback on our paper, which helped us improve its presentation and quality. We have carefully addressed all of your comments in the revised manuscript. We hope that you will be satisfied with the response provided by us.

Sincerely,

The authors.

 Editor's comments

Comment 1: Authors are requested to further enhance the proposed method with more techniques.

Response 1: The authors are thankful to the Editor for this comment in the revised version of the manuscript. We extended the paper's contributions by including a comparison between the strong classifier and the weak classifier (i.e., ensemble learning). Besides, we investigated the effect of combing the features extracted using a DL model with the SIFT model, a classic image descriptor. The list of contributions in the revised manuscript is as follows:

1. We proposed comparing deep transfer learning models' performance against using a hybrid classification method. In the proposed 

 hybrid classification method, the features are extracted using a proposed DL model. Then a classic machine learning classifier 

 classifies the chest CT scan images of the infected lung with COVID-19 and normal lung images.

2. We proposed utilizing ensemble learning (e.g., boosting and bagging methods) and a strong classifier (i.e., RF) for image 

 classification.

3. We proposed a new CNN model for feature extraction of COVID-19 lung CT scan images, which is trained from scratch. Then, we 

 proposed extracting the features using the well-known deep transfer learning models. The experimental results show that the quality 

 of the extracted features of the proposed CNN model is better than those features extracted by the well-known deep transfer learning 

 models.

4. The proposed methods are evaluated on five different metrics on one of the most used datasets. The obtained results show that the 

 proposed methods outperformed all existing state-of-the-art methods. Besides, the ensemble learning-based model slightly 

 outperformed the single classifier. 

Comment 2: Comparison against the state-of-the-arts should be done.

Response 2: The authors are thankful to the Editor for this comment. The original manuscript compares the proposed method against the top performing 12 state-of-the-art methods, as listed in Table 2.

 Reviewer #1's comments

The paper proposes a comparison between two approaches for predicting COVID-19 patients based on the CT scan images. The first approach uses deep transfer learning to classify the images, the second one is based on CNN to extract features from the images and classify them using ramdom forest algorithm.

Comment: The paper is well written and easy to understand. The problems being addressed are interesting. 

Response: The authors are thankful to the Editor for this comment.

Comment 1: However, the paper lacked novelty. The authors used deep learning techniques that are well known and widely used in medical image classification.

Response 1: The authors are thankful to the Editor for this comment. The main contribution of the proposed work was to answer the research question that whether to use the deep transfer learning, which is the current mainstream, or to design and train a proposed CNN model from scratch for the sake of detecting infected lungs with COVID-19 based on the CT scan images for feature extraction purpose only. The question is raised because the pre-trained models are trained on several types of images (e.g., car, building, and lung scan images.). In contrast, training a DL classifier model from scratch focuses on training the DL model on only lung CT scan images with normal and COVID-19 labels.

Thus, the novelty of this work is two-fold. First the proposed DL model, Fig. 2, for feature extraction. Second, answer the research mentioned above question. Besides, the proposed methods achieved the highest reported results for the widely used dataset.

Comment 2: I invite the authors to use more innovative methods and present in-depth the technical details.

Response 2: The authors are thankful to the Editor for this comment. As suggested, we extended the paper's contributions by including a comparison between the strong classifier and the weak classifier (i.e., ensemble learning). Besides, we investigated the effect of combing the features extracted using a DL model with the SIFT model, a classic image descriptor. Kindly check the list of contributions on the 2 and 3 pages at the end of the Introduction section.

---

## [Decision Letter · Decision Letter 1]

3 Aug 2022

PONE-D-21-38730R1A Hybrid CNN and Random Forest Model for COVID-19 Lung Infection Detection on Chest CT ScansPLOS ONE

Dear Dr. Hosny,

Thank you for submitting your manuscript to PLOS ONE. After careful consideration, we feel that it has merit but does not fully meet PLOS ONE’s publication criteria as it currently stands. Therefore, we invite you to submit a revised version of the manuscript that addresses the points raised during the review process.

We look forward to receiving your revised manuscript.

Kind regards,

Le Hoang Son, Ph.D

Academic Editor

PLOS ONE

**Comments to the Author**

1. If the authors have adequately addressed your comments raised in a previous round of review and you feel that this manuscript is now acceptable for publication, you may indicate that here to bypass the “Comments to the Author” section, enter your conflict of interest statement in the “Confidential to Editor” section, and submit your "Accept" recommendation.

Reviewer #2: (No Response)

Reviewer #3: (No Response)

2. Is the manuscript technically sound, and do the data support the conclusions?

Reviewer #2: (No Response)

Reviewer #3: Partly

3. Has the statistical analysis been performed appropriately and rigorously? 

Reviewer #2: (No Response)

Reviewer #3: Yes

4. Have the authors made all data underlying the findings in their manuscript fully available?

Reviewer #2: (No Response)

Reviewer #3: Yes

5. Is the manuscript presented in an intelligible fashion and written in standard English?

Reviewer #2: (No Response)

Reviewer #3: Yes

6. Review Comments to the Author

**Reviewer #2**: 

1) In introduction, introduce the problem, motivate the problem, and summarize the contributions

2) In literature survey, instead of summarizing the works, suggested to discuss the advatages and limitations

2) Authors are suggested to add detailed literature survey and some recommended and related works are

   [EDITORS HAVE REMOVED TO ENSURE BLINDNESS]

3) There are some typos and grmatical mistakes, one such typo is

Because we are dealing with a small dataset, we encouraged to use Scale-Invariant Feature Transform (SIFT) [?]

4) Add more details about the proposed architecture

5) Discuss the advatages and limitations of the proposed method

6) Compared the proposed approach with atleast recently published 3 methods

7) Add the level of trust of the proposed method

**Reviewer #3**: 

This paper presents a new CNN architecture for feature extraction and then to use the random forest (RF) for COVID-19 diagnosis.

The paper is well written and the presentation is adequate. However, the quality of the paper should be improved in parts.

Comments:

1)The author claims in the abstract that "To our knowledge, there is no research conducted to investigate whether to use the transfer deep learning models CT scan image classification against feature extraction."

I totally disagree on this since the following book chapter uses transfer learning models for deep feature extraction and classical ML methods for classification.

Abdulhamit Subasi, Arka Mitra, Fatih Ozyurt, Turker Tuncer, Automated Covid-19 detection from CT images us-ing deep learning in Editor(s): Varun Bajaj, G R Sinha, Computer-aided Diagnosis and Design Methods for Biomedical Applications, CRC Press, Taylor & Francis, 2021.

2)The novelty they claimed is not enough. Mayme they may say that their CNN model created from scratch achieved good result.

3)The title "A Hybrid CNN and Random Forest Model for COVID-19 Lung Infection Detection on Chest CT Scans" should be changed since it says Random forest, but RF did not achieved the best results after the revision, ensemble SVM achieved best accuracy.

Hence The title might be "A Hybrid CNN and Ensemble Model for COVID-19 Lung Infection Detection on Chest CT Scans"

3)What is the significant advantage of the proposed method against previously published ones? 

4)The literature review is not comprehensive. Authors are suggested to review more new and relevant research to support their research contribution.

---

## [Author Response · Author response to Decision Letter 1]

17 Sep 2022

Kindly find enclosed the response letter.

---

## [Decision Letter · Decision Letter 2]

3 Jan 2023

PONE-D-21-38730R2A Hybrid CNN and Ensemble Model for COVID-19 Lung Infection Detection on Chest CT ScansPLOS ONE

Dear Dr. Hosny,

Thank you for submitting your manuscript to PLOS ONE. After careful consideration, we feel that it has merit but does not fully meet PLOS ONE’s publication criteria as it currently stands. Therefore, we invite you to submit a revised version of the manuscript that addresses the points raised during the review process.

We look forward to receiving your revised manuscript.

Kind regards,

**Le Hoang Son, Ph.D**

Academic Editor

PLOS ONE

**Comments to the Author**

1. If the authors have adequately addressed your comments raised in a previous round of review and you feel that this manuscript is now acceptable for publication, you may indicate that here to bypass the “Comments to the Author” section, enter your conflict of interest statement in the “Confidential to Editor” section, and submit your "Accept" recommendation.

Reviewer #4: All comments have been addressed

2. Is the manuscript technically sound, and do the data support the conclusions?

Reviewer #4: Partly

3. Has the statistical analysis been performed appropriately and rigorously? 

Reviewer #4: Yes

4. Have the authors made all data underlying the findings in their manuscript fully available?

Reviewer #4: Yes

5. Is the manuscript presented in an intelligible fashion and written in standard English?

Reviewer #4: Yes

6. Review Comments to the Author

**Reviewer #4: **

1. What is the robustness of the proposed method?

2. Write about the advantages of the suggested method over other existing methods?

3. The originality of the paper needs to be stated clearly.

---

## [Editor Report · Decision Letter 3]

20 Feb 2023

A Hybrid CNN and Ensemble Model for COVID-19 Lung Infection Detection on Chest CT Scans

PONE-D-21-38730R3

Dear Dr. Hosny,

We’re pleased to inform you that your manuscript has been judged scientifically suitable for publication and will be formally accepted for publication once it meets all outstanding technical requirements.

Kind regards,

Le Hoang Son, Ph.D

Academic Editor

PLOS ONE

**Additional Editor Comments (optional):**

The paper has been sucessfully revised. It can be accepted in this form.

---

## [Editor Report · Acceptance letter]

27 Feb 2023

PONE-D-21-38730R3 

A Hybrid CNN and Ensemble Model for COVID-19 Lung Infection Detection on Chest CT Scans 

Dear Dr. Hosny:

I'm pleased to inform you that your manuscript has been deemed suitable for publication in PLOS ONE. Congratulations! Your manuscript is now with our production department. 

Kind regards, 

on behalf of

Prof. Le Hoang Son 

Academic Editor

PLOS ONE